# Manipulating multi-level selection in a fungal entomopathogen reveals social conflicts and a method for improving biocontrol traits

**Zoltan Erdos**[1]*, **David J. Studholme**[2], **Manmohan D. Sharma**[1], **David Chandler**[3], **Chris Bass**[1], **Ben Raymond**[1]*

1 Centre for Ecology and Conservation, University of Exeter, Penryn, United Kingdom, 2 Biosciences, University of Exeter, Exeter, United Kingdom, 3 School of Life Sciences, The University of Warwick, Coventry, United Kingdom

* z.erdos@exeter.ac.uk (ZE); b.raymond@exeter.ac.uk (BR)

**Data Availability Statement:** The genome assembly and annotation have been deposited at the GenBank database under accession number

## Abstract

Changes in parasite virulence are commonly expected to lead to trade-offs in other life history traits that can affect fitness. Understanding these trade-offs is particularly important if we want to manipulate the virulence of microbial biological control agents. Theoretically, selection across different spatial scales, i.e. between- and within-hosts, shapes these trade-offs. However, trade-offs are also dependent on parasite biology. Despite their applied importance the evolution of virulence in fungal parasites is poorly understood: virulence can be unstable in culture and commonly fails to increase in simple passage experiments. We hypothesized that manipulating selection intensity at different scales would reveal virulence trade-offs in a fungal pathogen of aphids, *Akanthomyces muscarius*. Starting with a genetically diverse stock we selected for speed of kill, parasite yield or infectivity by manipulating competition within and between hosts and between-populations of hosts over 7 rounds of infection. We characterized ancestral and evolved lineages by whole genome sequencing and by measuring virulence, growth rate, sporulation and fitness. While several lineages showed increases in virulence, we saw none of the trade-offs commonly found in obligately-killing parasites. Phenotypically similar lineages within treatments often shared multiple single-nucleotide variants, indicating strong convergent evolution. The most dramatic phenotypic changes were in timing of sporulation and spore production *in vitro*. We found that early sporulation led to reduced competitive fitness but could increase yield of spores on media, a trade-off characteristic of social conflict. Notably, the selection regime with strongest between-population competition and lowest genetic diversity produced the most consistent shift to early sporulation, as predicted by social evolution theory. Multi-level selection therefore revealed social interactions novel to fungi and showed that these biocontrol agents have the genomic flexibility to improve multiple traits—virulence and spore production—that are often in conflict in other parasites.

JAJHUN000000000. The version used in this paper is JAJHUN010000000. The raw sequence data including Illumina libraries for evolved lineages can be found under BioProject accession PRJNA777543. Phenotypic data available at: Erdos, Zoltan (2024): Akanthomyces muscarius multi-level selection [Dataset]. Dryad. https://doi.org/10.5061/dryad.9s4mw6mpz.

**Funding:** This study was supported by an Agriculture and Horticulture Development Board (AHDB) studentship award (CP 176) to BR and by BBSRC award BB/S002928/1 to BR. The funders did not play any role in the study design, data collection and analysis, decision to publish, or preparation of the manuscript.

**Competing interests:** The authors have declared that no competing interests exist.

## Author summary

Understanding the ecological forces that shape virulence is a key challenge in evolutionary biology. Here we investigated how competition at different levels of selection (within-hosts, between-hosts, between populations) could alter investment in virulence in a fungal entomopathogen. We predicted that cooperative investment in virulence would increase at higher scales of competition and aimed to further our understanding of potential trade-offs shaping life-history of a fungal insect pathogens. We found moderate increases in virulence in different selection regimes. Importantly, we did not find commonly expected trade-offs, such as that between spore production and virulence or a trade-off between virulence and growth rate that is consistent with cooperation. However, we found convergent genetic changes and significant differences in timing and production of spores, dependent on how we manipulated scales of selection. Our data suggests that this is driven by social conflict regarding the timing of sporulation. This carries fundamental importance for understanding how varying selection pressure at different scales shape pathogen life history. In addition, these results also have applied importance for understanding how to improve and select for beneficial traits in biocontrol agents.

## 1. Introduction

Understanding how parasite biology and ecology shapes the evolution of virulence is an open question of both fundamental and applied importance. This question is of particular relevance for microbial biological control agents as researchers have commonly attempted to manipulate the virulence and host range of potential biological control agents through experimental evolution [1]. A range of parasite life history traits covary with virulence. These covariations are sometimes described as trade-offs, such as that between transmission rate and virulence [2,3] while others are described as evolutionary conflicts, such as the relationship between replication rate and virulence in social microbes [4,5]. The biology of different parasites shapes which of these trade-offs are most important in each case.

A starting point for the discussion of the evolution of virulence are 'classical' theories, which focus on how selection affects virulence in terms of the reduction in fitness of hosts during infection. A common outcome of these theories is that optimal virulence is often at an intermediate level, balancing the need to extract resources from hosts with fitness effects that can curtail duration or efficacy of transmission [6,7]. From the perspective of selection at the within-host scale, it is important that virulence in classical models is positively related to parasite growth rates. For entomopathogens that are obligate killers such as the biocontrol fungus *Akanthomyces muscarius*—the organism used in this study–the death of hosts is essential for transmission, so the normal trade-offs, in terms of predicting how much harm is caused by infection, do not apply [8]. Nevertheless, there can still be trade-offs in terms of the *timing* of host death as too rapid death can limit pathogen propagule production, termed yield: a trade-off between rapid killing and effective propagule production is widespread in invertebrate pathogens [9–15]. There are two mechanisms at work here: increasing speed of kill can reduce pathogen yield in invertebrates as hosts may be feeding and growing during infection [16]. In addition, higher virulence may have trade-offs in terms of pathogen propagule production that are independent of host size, because additional resources are required for investment in virulence and are diverted away from reproduction: this trade-off can affect propagule production *in vitro* or *in vivo* [17–19].

Passage of invertebrate pathogens is a commonly used method to restore or maintain virulence. However, distinction should be made between a single passage to restore virulence of a potentially attenuated strain and serial passage to increase virulence via selection on allelic variation. Rapid recovery of traits may be related to DNA methylation [20]. In terms of artificial selection, the published evidence on the efficacy of passage on selecting for virulent fungal variants is mixed at best [1]. Several studies that aimed to increase fungal virulence via passage have failed to do so [21–23]. There are some successful examples of serial passage increasing fungal virulence [24,25], in one of these cases increased virulence was a result of simple passage through the host [24]. These results show that in principle it is possible to increase the virulence of entomopathogenic fungi (EPF) via artificial selection, but success is often poor in practice and selection designs have not been informed by theory, which could explain the conflicting results seen to date.

Furthermore, we often have a poor understanding of the selection pressures maintaining a wide range of parasite life history traits, not just virulence. This means that essential traits can be difficult to maintain in the laboratory, even when using *in vivo* propagation in a model host. Examples of this include virulence in entomopathogenic nematodes [26–28] and sporulation in entomopathogenic fungi [20]. For example, multiple studies using *in vivo* passage of fungal pathogens in insect hosts have observed reduced spore production following selection [25,29]. Maintaining efficient sporulation is essential for the development of successful formulations used in biological control [30]. Sporulation and virulence are therefore key traits for commercial exploitation of insect pathogens but it is not always clear how to design passage protocols to maintain these complex traits.

For invertebrate pathogens, social evolution theory e.g. cheat-cooperator conflict, has been particularly valuable in terms of understanding the evolution of virulence [5,17–19,31]. This can take the form of conflicts over investment in public goods such as the Cry toxins produced by the bacterium *Bacillus thuringiensis* [17]. These proteins are extremely costly for individual cells to produce but are solubilized during infection and provide benefits to groups of cells invading hosts [17]. Cheats in this context are cells that gain a growth advantage in mixed infections by avoiding the cost of producing public goods [32]. This theory has relevance beyond bacteria as cheat invasion provides a good explanation for the instability of virulence in entomopathogenic nematodes [33]. Social evolution theory is not necessarily in conflict with classical models of virulence and can be embedded within this framework [34]. However, classical models of virulence and some forms of cooperative virulence (e.g. public goods production) do make contrasting predictions about how pathogen life history correlates with virulence. In classical models growth within hosts positively correlates with virulence; in public goods models, virulence is costly, so that in mixed infections producers that invest in virulence have slow growth rate relative to mutants that do not produce virulence factors [1].

A common feature of all evolutionary models of virulence is that selection across multiple scales–such as within-host growth and between-host transmission–shapes the overall fitness of pathogen genotypes. Altering the strength of selection at different scales can therefore have important consequences. For example, increasing within-host competition can select for rapid growth rate [35]. As discussed above rapid growth can increase or decrease virulence depending on whether pathogens fit classical virulence or public goods frameworks [36–40]. Between-host selection, on the other hand, which acts on yield or population size, is expected to maintain investment in group beneficial traits such as iron-scavenging siderophores [4,41]. Finally, for virulence traits such as the Cry toxins of *B. thuringiensis*, competition between-populations of infected hosts maintains investment in virulence most effectively, as these toxins impose costs on both pathogen yield and growth rate [19]. It is therefore important to

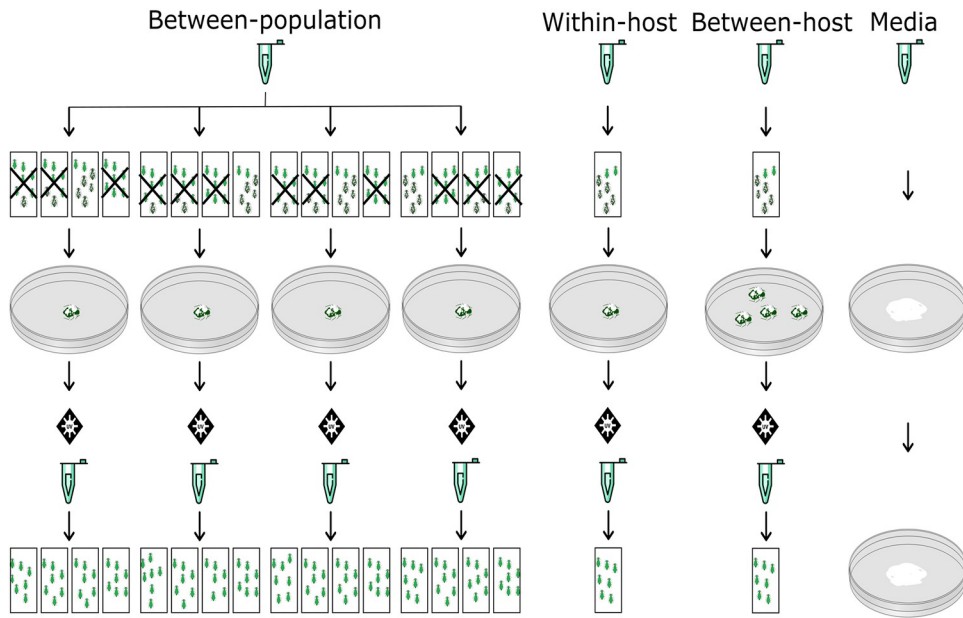

**Fig 1. Selective passage protocol: Rectangles represent individual bioassay chambers containing 11–13 adult apterous aphids on a single leaf of Chinese cabbage.** Between-population treatment: The first mycosed cadaver from the subpopulation with highest recorded mortality was used to propagate spores for the next passage. Crossed out bioassay chambers were discarded from the next round of passage (lineages belonging to treatment: BPI 1–4). Within-host treatment: The first mycosed cadaver was used to propagate the next round of infection (WHS 1,3,4,8). Between-host treatment: The first 3–4 mycosed cadavers were pooled, homogenized and plated to propagate the next round of infection (BHY 3,5). Media treatment: mycelial growth taken from an SDA plate is taken with a cork borer and placed on a new SDA plate (media 1–3).

consider selection at multiple scales, including that above a single infection, when studying the evolution of virulence.

In this study we hypothesized that altering the intensity of selection at different scales would have varied impacts on fungal pathogen virulence and reveal trade-offs in fundamental life history traits. We used the entomopathogenic fungus *A. muscarius* in our experiments with *Myzus persicae*, the peach potato aphid, our model host. We chose to develop this experimental system because there are few microbial control solutions for hemipteran pests, and understanding how to improve or select for virulence in aphid pathogens is of interest in pest management. *Akanthomyces* is a genus of entomopathogenic fungi that contains genotypes that are adapted to causing infection in hemipteran insects, and one isolate of *A. muscarius* has been sold since the 1980s as the biocontrol agent "Mycotal" (Koppert BV, Netherlands) primarily for use in protected crops [42,43]. The fungal infection cycle involves spore adhesion, germination, differentiation of infection structures, penetration of cuticle, colonization of the hemocoel and sporulation following emergence from the mycosed cadaver. The development of commercial entomopathogenic fungal control agents is based around identifying 'winning' isolates with desirable traits from a candidate pool [44]. However, to the best of our knowledge, there is no action taken to improve or maintain the activity of chosen isolates through artificial selection grounded in evolution theory.

By manipulating intensity of selection at different scales (between-populations, between-hosts, within-hosts) we sought to impose selection for life-history traits (infectivity, yield, speed of kill) important in biocontrol (Fig 1). Since fungal entomopathogens use a wide range of secreted metabolites during infection [45,46], which we hypothesized would be costly to

produce, we predicted that direct selection for infectivity via competition between-populations of infected hosts would provide the most effective selection for virulence, as seen previously for bacterial social traits [19]. We hypothesized that by pooling multiple aphid cadavers (increased between-host competition) we would select for increased pathogen propagule production (yield). Finally, stronger within-host competition was predicted to lead to more rapid growth and potentially cheating strategies. On the one hand, rapid growth could result in a decrease in virulence if public goods or cooperative investment is involved in host infection. However, classical virulence theory makes a contrasting prediction here and predicts that rapid growth within host could result in earlier death with a potential trade-off in pathogen yield.

Starting with a genetically diverse pathogen we imposed selection over 7 rounds of selection (passages) and used UV mutagenesis every other selection round. We genetically and phenotypically characterized evolved lineages and the ancestral stock by whole genome sequencing and measuring changes in life history (virulence, linear growth rate, sporulation). While several lineages showed small but significant increases in virulence, there was no evidence for any of the trade-offs commonly seen in obligately-killing parasites. Phenotypically similar lineages within treatments commonly share multiple SNPs indicating strong convergent evolution in this pathogen. The most dramatic change was apparent in the timing and production of spores (conidia) *in vitro*. The between-population selection regime produced earliest sporulation and highest yield of spores as well as the fewest numbers of genetic variants within evolved lineages, suggesting that this regime might be the most efficient at purging genetic variants and minimizing social conflicts on timing of sporulation. In line with social evolution predictions, evolved lineages with earlier sporulation showed reduced competitive fitness. While we did not see strong predicted trade-offs in terms of evolution of virulence, timing of sporulation did respond to selection treatments designed to minimize within group conflict and provide group level benefits.

## 2. Results

We hypothesized that the between-population (BP) treatment would result in strongest selection on infectivity, that the between-host (BHY) treatment would most efficiently select for increased pathogen yield and the within-host treatment (WHS) would either increase speed of kill or lead to cheating and loss of virulence. We carried out seven rounds of passage with *A. muscarius*, although all treatments received a standard inoculum dilution at each round of infection, replicate level mortality varied between 25 and 80% across the experiment as a whole.

Not all replicates of the within-host and between-host treatments survived until the final seventh round of passage due to extinction events. We anticipated that increased within-host competition might lead to more cheating. Four replicates went extinct in the within-host treatment because inocula failed to cause successful infections within 7 days, which is consistent with loss of essential virulence or sporulation traits via cheat invasion. Six replicates of the between-host treatment were lost due to bacterial contamination following the pooling and homogenization of aphid cadavers (Fig 2A).

### Virulence of artificially selected *A. muscarius* towards *M. persicae* and *B. brassicae*: infectivity and time to kill

Fungal bioassays with the ancestral and evolved lineages tested how overall virulence and different components of virulence (total mortality and time to death) varied with selection

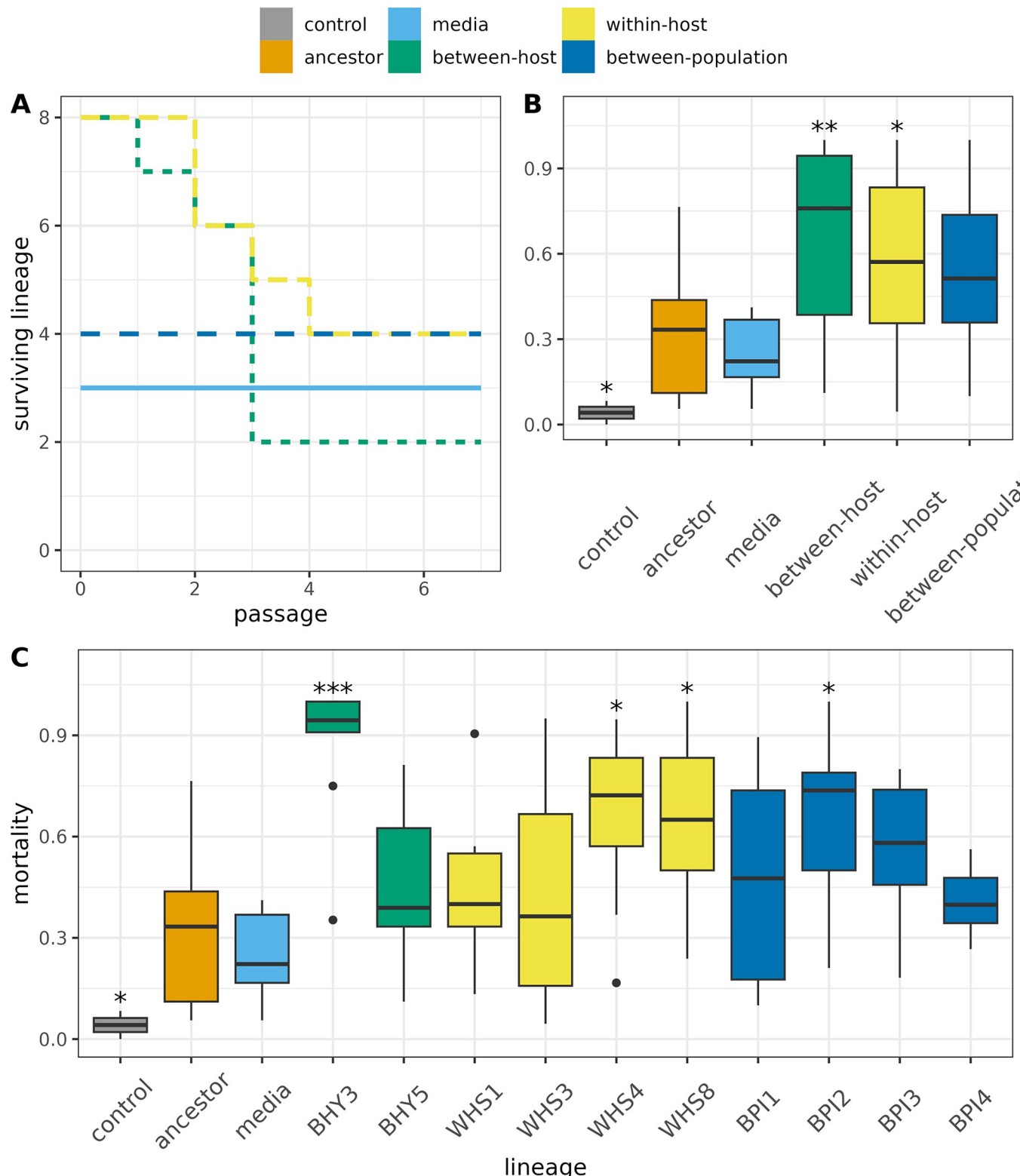

**Fig 2.** Surviving lineages within each treatment showing extinction of between-host and within-host replicates at different passages (A). Total mortality of M. persicae in bioassays following 7 days of exposure to 3x10$^6$ conidia/ml of spore preparations of ancestral and evolved lineages after 7 rounds of selection. Plot B represent treatment level mortality while plot C shows variations between independently evolved lineages within treatments. Significance levels denoted by asterisk are pairwise comparisons to the ancestor using t-tests: ***: p < = 0.001; **: p < = 0.01; *: p < = 0.05 Boxplots showing medians, first and third quartiles, whiskers are 1.5 * interquartile range (IQR). Data beyond the end of the whiskers are outliers.

treatments designed to alter these life history traits. Since we confirmed fungal infection (mycosis) for all cadavers, mortality is equivalent to infectivity in this obligately-killing pathogen.

Overall, bioassays were robust with very low mortality in all pathogen-free controls (3.03% ± 2.9 after 7 days). There was a clear minimum time to death as fungal treatment increased aphid mortality on day four and onwards ($\chi^2$ = 30.5, $df$ = 5, p < 0.001). We examined changes in overall virulence with survivorship analysis (likelihood ratio = 159.5, df = 5, p < 0.001). This analysis predicted significantly higher hazard ratios for the within-host (1.84, 95% CI [1.38, 2.45]), between-host (2.94, 95% CI [2.17, 3.96]) and between-population (1.78, 95% CI [1.33, 2.38]) treatments compared to the ancestor [1].

Changes in overall virulence were predominantly related to total mortality rather than the timing of death although phenotypic effect sizes were modest: the largest changes were in lineages in the between-host treatment with 2-fold increase in total mortality relative to the ancestor. As with overall virulence, the within-host and between-host treatments produced significant increase in total mortality over the 7-day assay, relative to the ancestor ($\chi^2$ = 57.6, $df$ = 5, p < 0.001, Fig 2B).

Response to selection in terms of virulence was heterogeneous within all treatments- not all lineages increased in virulence. Survivorship analysis indicated that selected lineages BPI2, BPI3, WHS4, WHS8 and BHY3 had a significantly higher instantaneous risk of death (likelihood ratio = 301, df = 12, p < 0.001 Fig 2C). Lineages BPI2, BPI3 from the between-population treatment, WHS4, WHS8 from the within-host treatment and BHY3 from the between-host treatment achieved significantly higher mortality in aphids after 7 days of exposure ($\chi^2$ = 83.98, $df$ = 12, p < 0.001, Fig 2C).

Since we made specific predictions about speed of kill we explored changes in this component of virulence separately. Here, we found that few lineages showed any response to selection. There was variation in speed of kill ($F_{11, 937}$ = 8.93, p < 0.001, S1 Fig) but changes were mostly restricted to lineage BHY3 (mean time to death 5.73 days) and WHS1 (6.64 days) compared to the ancestor (6.22 days). Model simplification (pooling all treatments except media and between-host does not prompt a significant loss of deviance) suggests that this effect is predominantly driven by lineage BHY3 ($F_{2, 946}$ = 28.46, p < 0.001).

Virulence of a subset of the evolved lineages were tested against *B. brassicae*, a different host to the one used for selection, to test if patterns of increased virulence are host specific or not. Fungal treatment had a significant effect on *B. brassicae* mortality on day 3 ($\chi^2$ = 15.8, $df$ = 4, p < 0.001) and onwards. Mean control mortality after 7 days was 13.7% ± 4.8. No difference was found between the total mortality of the ancestor and the evolved lineages ($\chi^2$ = 2.7, $df$ = 3, p = 0.44, S2A Fig). However, survivorship analysis revealed a significantly higher instantaneous risk of death for the between-population lineage BPI2 (HR = 1.58, 95% CI [1.22, 2.04]) and the between-host selected lineage BHY3 (HR = 1.50, 95% CI [1.07, 2.12]) compared to the ancestor [1] (likelihood ratio = 92.6, df = 4, p < 0.001, S2B Fig).

## Spore production and timing of sporulation

We hypothesized that changes in virulence or timing of death might have trade-offs in terms of spore productivity *in vitro* or in timing of sporulation. We tested how selection regimes affected both early sporulation (3 days post inoculation) and the total production of conidia after two weeks of culture on solid media. In general, selection treatment, rather than the lineage specific changes in virulence, shaped spore production phenotypes. This suggests that variation in the intensity of selection at different levels was particularly important for this trait. Selection treatment had significant effects on early sporulation ($\chi^2$ = 20.33, $df$ = 4, p < 0.001).

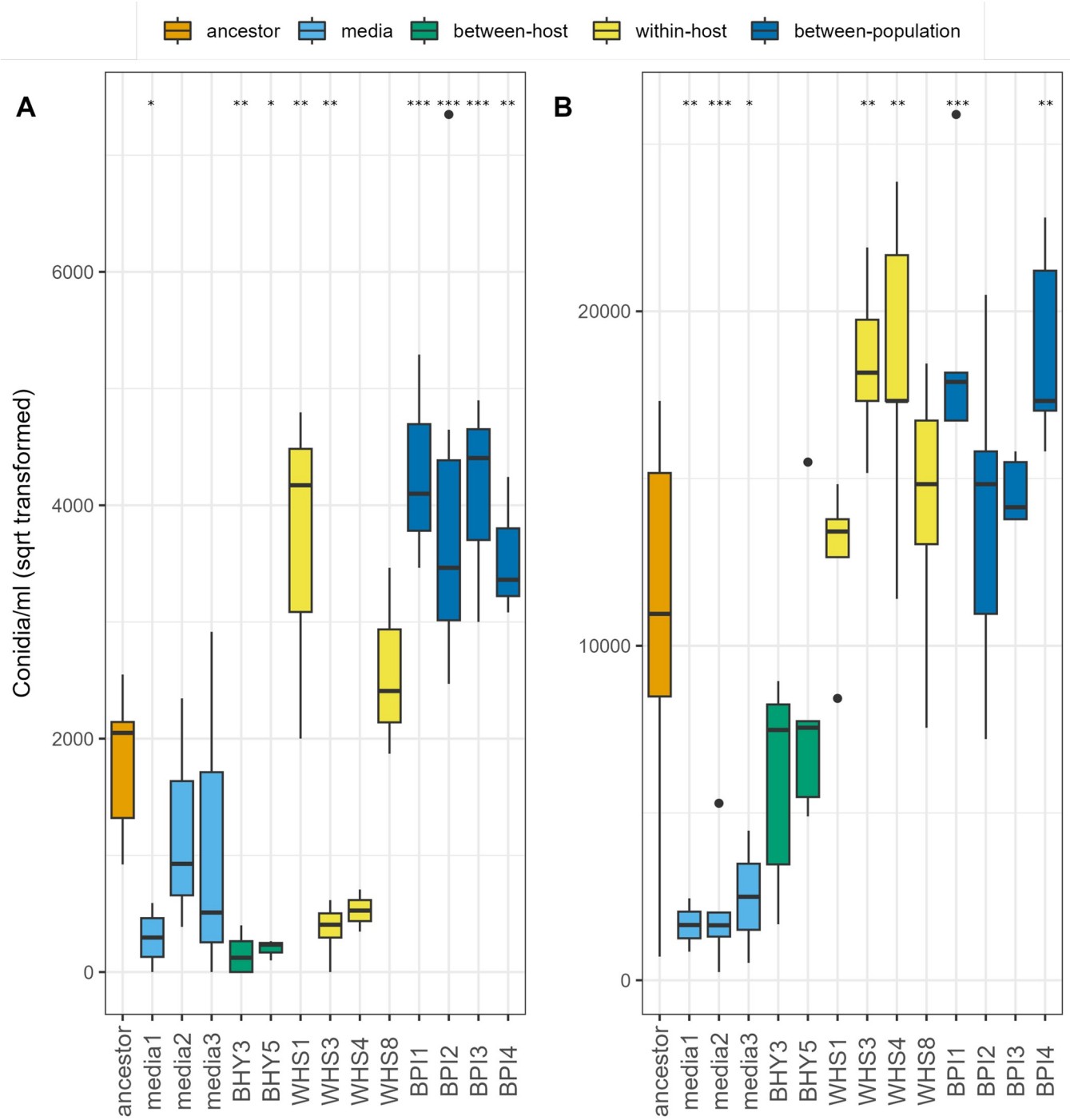

**Fig 3.** Spore production of ancestor and evolved fungal lineages after 3 days (A) and 14 days (B). Boxplots showing median, first and third quartiles, whiskers are 1.5 * interquartile range (IQR). Data beyond the end of the whiskers are outliers. Significance levels denoted by asterisk are planned pairwise comparisons to the ancestor using T-tests: **: p < = 0.01; *: p < = 0.05.

Both surviving between-host treatment lineages (BHY3 and BHY5) had low early sporulation relative to the ancestor (Fig 3A), despite having very different virulence characteristics- one had high virulence and low sporulation while the second had low virulence and low

sporulation- showing no evidence of a consistent trade-off. All four lineages belonging to the between-population treatment showed significantly higher early sporulation, while only one of these lineages (BPI2) had a strong increase in virulence. The within-host selection treatment resulted in a mixed outcome with lineages WHS1 producing significantly more spores ($1.3 \times 10^7$ conidia/ml) whereas WHS3 produced fewer spores ($1.4 \times 10^5$ conidia/ml). Spore production was not statistically different between lineages media1, media3, WHS4, WHS8 and the ancestor ($3.2 \times 10^6$ conidia/ml) ($F_{13, 43} = 13.15$, p < 0.001, Fig 3A).

The effect of selection treatment on final sporulation, i.e., after 14 days of growth was also clear ($\chi^2 = 31.02$, $df = 4$, p < 0.001). All lineages selected *in vitro* (media 1, 2 and 3) showed significantly lower sporulation, two lineages in the between-population treatment (BPI1 and BPI4) and two lineages in the within-host treatment (WHS3 and WHS4) produced significantly more spores compared to the ancestor ($F_{13, 50} = 11.28$, p < 0.001, Fig 3B).

## Growth rate and relative fitness

If increases in virulence are the result of increased investment in public goods, then high virulence should be accompanied by reduced growth rate. We use radial growth of colonies as a proxy for growth rate in all evolved lineages, as is common in fungal life history studies. The different selection treatments had a significant effect on mycelial growth at 22°C *in vitro*, although effect sizes were small ($F_{14, 1209} = 4878$, p < 0.001, Fig 4). As with the sporulation life history data, changes in radial growth rate did not correlate with changes in virulence. Two of the lineages with increased virulence, BHY3 and WHS4, did show reduced radial growth relative to the ancestor, but this effect was not consistently related to changes in virulence. For example, two strains from the between-population treatment, BPI1 and BPI4, had a faster growth rate than the ancestor; WHS3 also had reduced radial growth despite no change in virulence. The between-host selection treatment resulted in slower growth rate for both lineages BHY3 and BHY5.

Although radial growth *in vitro* is widely used as a proxy for fungal growth rate, it has limitations as a measure of total biomass since fungal colonies have a three-dimensional structure on agar plates. In this instance it is more relevant and accurate to measure relative growth using competition assays. Since fungi lack readily scorable phenotype traits that can be used to distinguish competitors, this is not a trivial process. Using resequencing of evolved lineages and the genome of the ancestral pool, we designed qPCR probes on a polymorphic allele to distinguish the ancestor and three lineages from the between-population treatment from a common competitor passaged on media.

We focused on the lineages belonging to the between-population selection treatment as these showed the strongest phenotypic response in sporulation traits under selection. Since this between-population competition is designed to minimize selection and maximize the benefits associated with group beneficial traits, we hypothesized that early sporulation is a social trait that results in low fitness in competition. As we hypothesized, the relative fitness of the evolved lines belonging to the between-population treatment were significantly lower than the ancestor ($F_{3,8} = 18.2$, p < 0.001, Fig 5A) while reductions in fitness were strongest in evolved lineages showing strongest early sporulation (Fig 5B).

## Variant calling and convergent evolution

We re-sequenced all evolved lineages with the aim of identifying patterns of shared evolutionary change in independent lineages that are characteristic of convergent evolution. In addition, we characterized the standing genetic variation in our ancestor strain and the effect of selection on number of genetic variants in each evolved lineage.

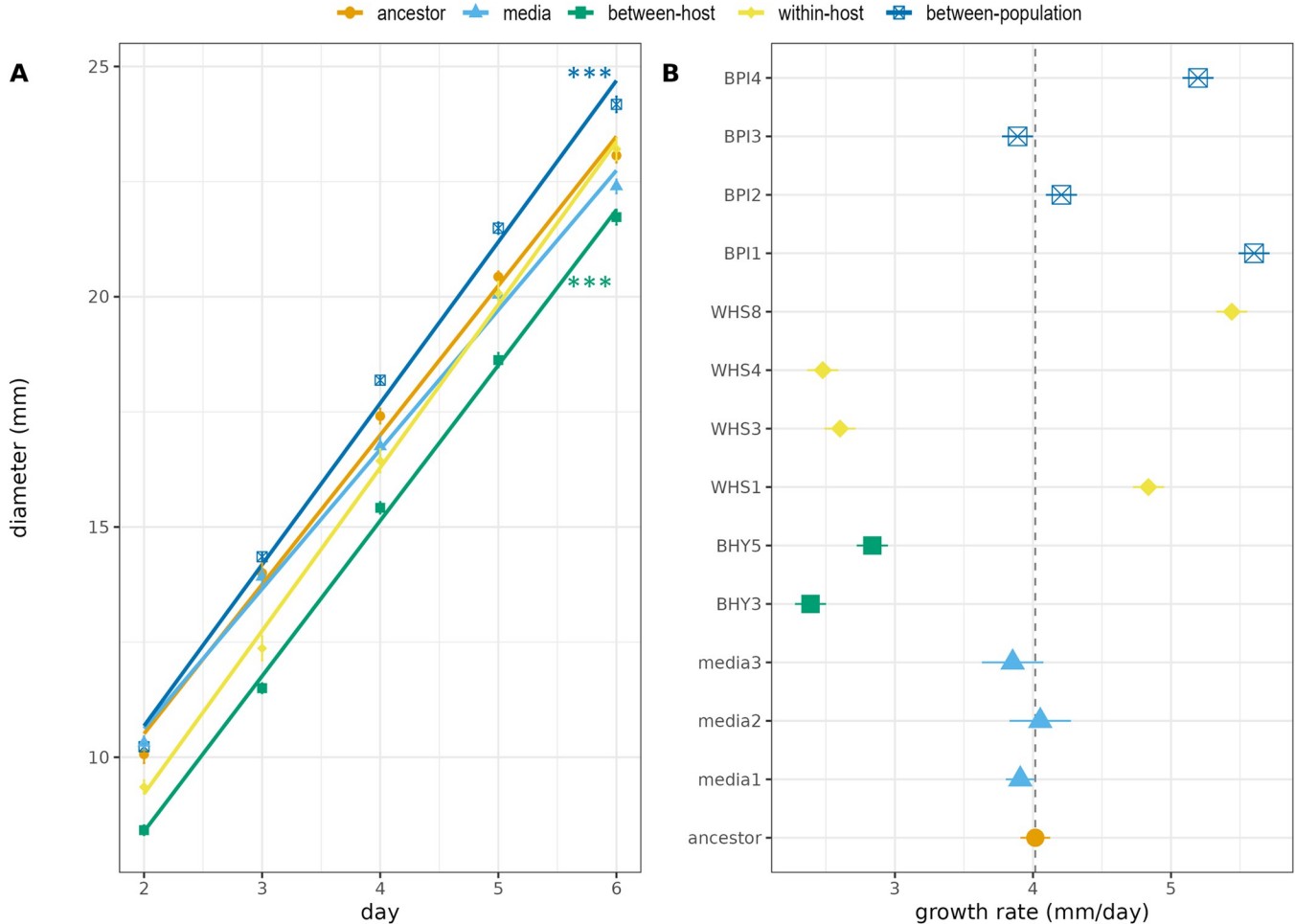

**Fig 4.** Growth curves of selected lines of A. muscarius. Significance levels denote marginal means comparisons to ancestor using Tukey's HSD. *** = p < 0.001. (A). Linear model estimates for growth rate of individual replicates belonging to selection treatments (B).

We identified 205 high-confidence variant sites within the 14 samples. These comprised 58 insertion or deletions (indels) and 147 single nucleotide polymorphisms (SNPs). The ancestral population already carried genetic variation at 50 sites, whereas 155 sites of genetic variation were unique to evolved lineages (Fig 6).

Out of the total number of variants that passed filtering, 45 SNPs and 18 indels show convergent evolution, *i.e.* were present in at least half of the replicates within each selection treatment. The highest number of unique single-nucleotide variants showing convergent evolution were found in the within-host selection treatment [14] followed by the between-host [9] and between-population [5] treatments. We identified only one SNP that diverged from the ancestral genotype in more than half of the replicates in the media treatment. Sequence similarity searches against the NCBI nr database identified matches to previously described protein coding sequences (Table 1). These sequences include those encoding Heterokaryon Incompatibility protein, with a putative role in vegetative and sexual recognition systems during vegetative growth [47]. Also identified were, Transcription Elongation Factor Eaf that has been linked with an effect on long transcript synthesis in *Drosophila* [48], a Phosphatidate Phosphatase (PAP) enzyme that regulate the synthesis of phospholipids and triacylglycerol in yeast [49], ATP-binding cassette (ABC) transporter involved in substrate translocation across membrane

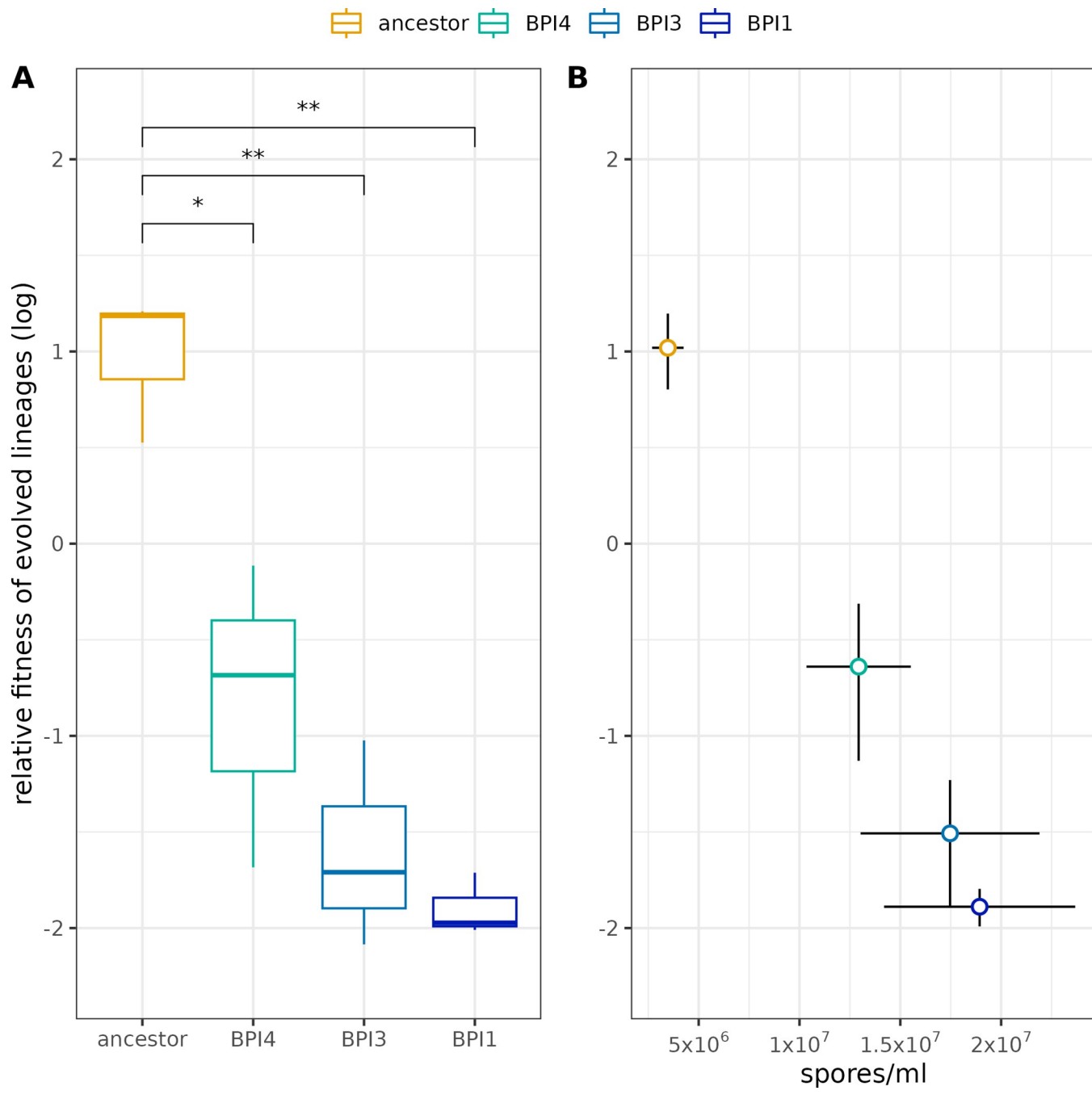

**Fig 5. Relative fitness of ancestral and evolved lineages of the between-population treatment in competition with media1 lineage.** Relative fitness values are derived from relative fluorescence unit measurements from TaqMan fluorescence data and are log transformed. Significance levels denoted by asterisk are pairwise comparisons to the ancestor using unpaired two samples t-test: **: $p \leq 0.01$; *: $p \leq 0.05$ (A). Relative fitness plotted against early spore production of each lineage (B). Points represent the mean, lines are standard errors of the mean.

[50,51], a RING finger (RNF) protein with a role in ubiquitination and DNA repair [52], F-actin capping protein that restricts actin filament growth in Drosophila [53], FAD-dependent oxidoreductase catalyzing glucose oxidation [54], Cyclic Nucleotide Phosphodiesterase (PDE) that belongs to a group of enzymes degrading the phosphodiester bond in the second

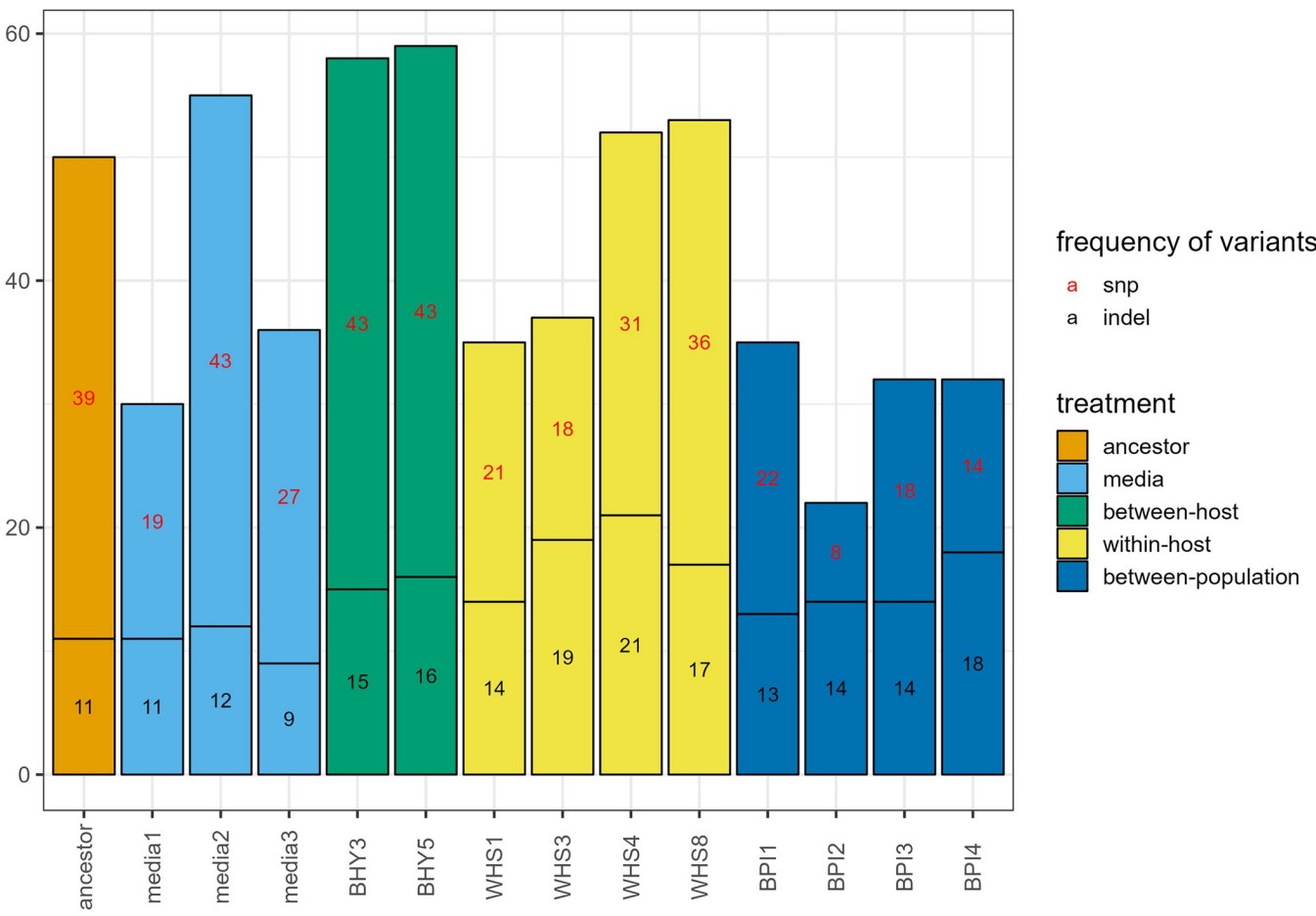

**Fig 6. Number of genomic variants per fungal lineage.** Colours represent different selection treatments.

messenger molecules cAMP and cGMP [55]. The search also matched the sequence of Methionine Permease involved in transmembrane activity [56].

### Integration with phenotypic data

As mentioned above, our variant discovery highlights multiple unique genomic variants that show signs of convergent genomic evolution within selection treatments. Furthermore, these genomic variants overlap with some of the phenotypic clusters belonging to different selection treatments (Fig 7) indicating that genotypic convergence is matched by phenotypic convergent evolution in several cases. These convergent variants are spread across 9 contigs representing different chromosomes.

### 3. Discussion

Our hypothesis was that fungal virulence, being partly based on the action of secreted metabolites, would follow the model of public good virulence [5,57]. In line with social evolution theory, we expected that selection treatments that maximized within-host competition might lead to cheating and loss of virulence, while selection treatments that optimized group level benefits by selecting for infectivity between populations of infected hosts would provide the most efficient means of increasing virulence [1]. While a number of selected lineages showed modest

**Table 1. Regions containing convergent variants and their best respective blast hits.** All mutations were located within coding regions in putative ORFs. Ref: reference allele, Alt: alternate allele, E-value: the number of alignments expected by chance with the calculated score or better, % iden: the highest percent identity for a set of aligned segments to the same subject sequence, accession: NCBI accession number, treatment.

| contig | position | Ref. | Alt. | type | description | scientific name | E -value | %. iden | accession | treatment |
|---|---|---|---|---|---|---|---|---|---|---|
| 3 | 2408047 | A | | INDEL | Transcription elongation factor Eaf | *Beauveria brongniartii* RCEF 3172 | 7.00E-149 | 93.51 | OAA75250.1 | BP |
| 18 | 1502511 | G | | INDEL | actin cytoskeleton organization protein APP1 | *Akanthomyces lecanii* RCEF 1005 | 0 | 98.54 | OAA74498.1 | BP |
| 19 | 2107579 | A | G | SNP | ABC drug exporter AtrF | *Akanthomyces lecanii* RCEF 1005 | 0 | 99.87 | OAA76511.1 | BP |
| 20 | 2512846 | G | A | SNP | RING finger domain containing protein | *Cordyceps militaris* | 0 | 92.46 | ATY64=3212.1 | BP |
| 20 | 4112482 | G | | INDEL | F-actin capping protein subunit | *Beauveria bassiana* | 2.00E-118 | 87.42 | PMB73233.1 | BP |
| 4 | 2881225 | C | A | SNP | Heterokaryon incompatibility | *Akanthomyces lecanii* RCEF 1005 | 0 | 98.52 | OAA77840.1 | WHS |
| 8 | 1458577 | T | A | SNP | FAD dependent oxidoreductase | *Akanthomyces lecanii* RCEF 1005 | 6.00E-133 | 84.16 | TQV96894.1 | WHS |
| 18 | 1176621 | T | † | INDEL | 3'5'-cyclic nucleotide phosphodiesterase | *Akanthomyces lecanii* RCEF 1005 | 0 | 91.52 | OAA74610.1 | WHS |
| 18 | 2006362 | GCT | G | INDEL | high affinity methionine permease | *Akanthomyces lecanii* RCEF 1005 | 2.00E-151 | 99.57 | OAA74361.1 | WHS |
| 2 | 2759322 | T | C | SNP | Transcription factor | *Akanthomyces lecanii* RCEF 1005 | 0 | 96.62 | OAA72753.1 | BHY |
| 16 | 911838 | T | A | SNP | Fungal specific transcription factor | *Akanthomyces lecanii* RCEF 1005 | 2.00E-80 | 93.39 | OAA71766.1 | BHY |

increases in virulence, we did not see evidence of the expected trade-offs in terms of either altered growth rate or sporulation efficiency and so no evidence for public goods based virulence.

Instead, the most robust phenotypic changes in response to experimental evolution were in traits related to timing and efficiency of sporulation. These were also most consistently linked to selection treatments. Notably the between-population treatment clearly produced earlier sporulation. In contrast, selection of control lines on media only produced lineages with late, inefficient sporulation. Although between-population selection was designed to maximize virulence (infectivity per spore), it was possible in our experimental design for evolving lineages to alter infectivity by increasing the efficiency of conidia production, and our data are consistent with this interpretation. Sporulation, or more generally the formation of viable non-replicating cells, also has the potential to be a cooperative behavioural trait in microbes [58,59]. The rationale here is that the switch to a dormant resting state has consequences for neighbouring microbes: it is a switch away from growth that can free up resources for actively dividing cells that have not yet committed to sporulation. Current social evolution theory has primarily considered non-sporulating mutants rather than variation in the timing of sporulation [58,60]. However, the same concept of social conflict applies in that early sporulating cells are in danger of being out-competed by later sporulators in mixed culture. Ultimately these conflicts can lead to a 'tragedy of the commons' as cells that sporulate late may not reserve optimal levels of resource and endanger their progeny with starvation or dehydration. This study shows clear evidence of this tragedy of the commons: lineages that sporulate earlier have low competitive fitness against later sporulating microbes, but when cultured alone can produce spores more efficiently. A caveat here is that we observed multiple mutations in lineages with early sporulation–changes in spore productivity and competitive fitness may not be directly linked to this

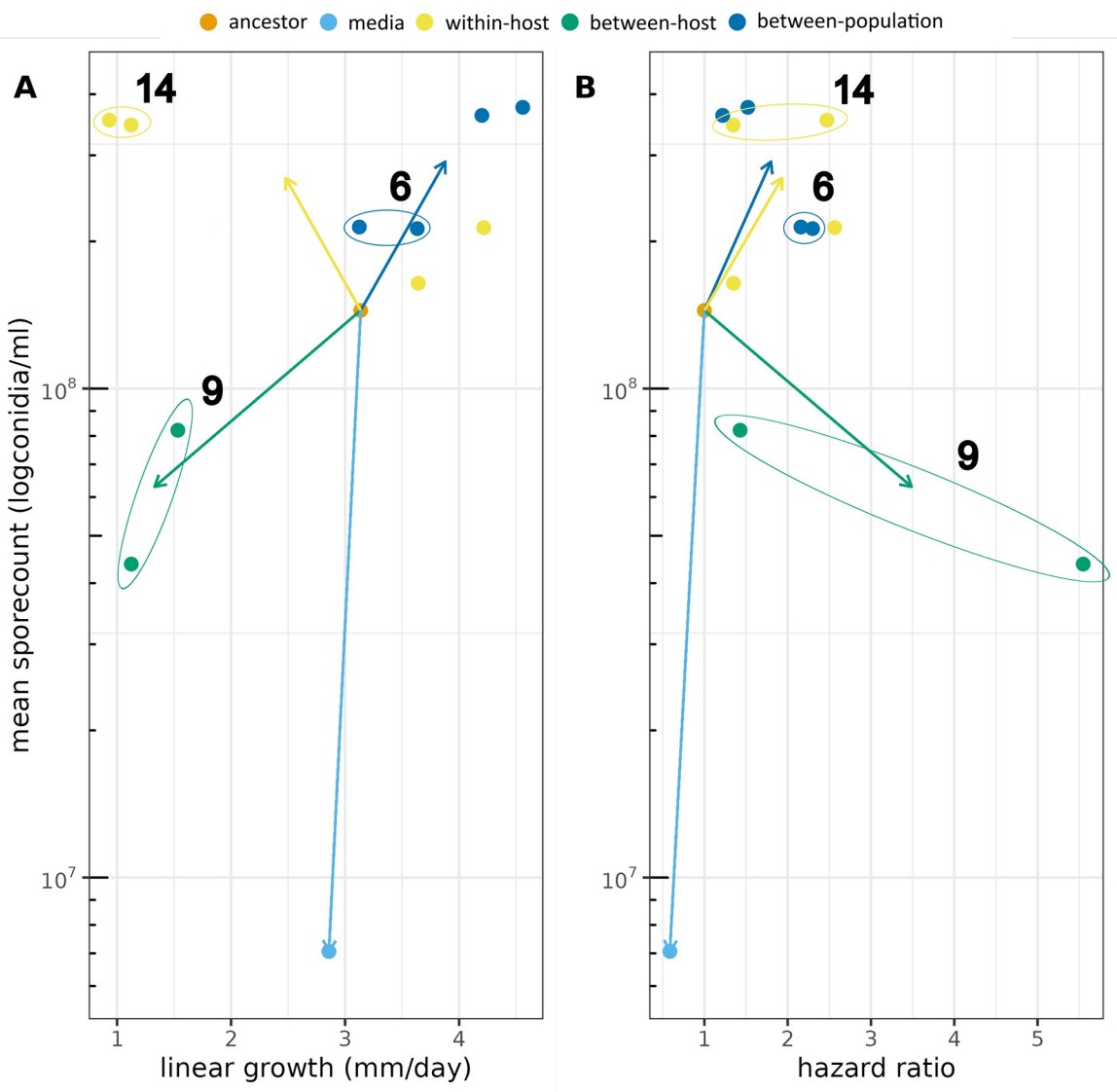

**Fig 7. Trends in phenotypic space of individually evolving fungal lineages belonging to different selection treatments.** Mean spore count is on a $\log_{10}$ scale. Points represent means per individually evolving replicate. Arrows lead from ancestor to mean of selected lineages. Virulence is represented as the estimate of instantaneous hazard parameter from survivorship analyses. Encircled clusters are colour coded according to treatment with a number representing the number of unique genomic variants identified belonging to the cluster.

trait. Screening this panel of SNPs and recreating these mutations in controlled background would be a major technical challenge for a non-model species and is beyond the scope of this study. A future consideration is how the possible conflict on timing of sporulation will play out in natural infections, rather than on agar plates. EPF sporulate on the outside of the host and early sporulation may also translate to earlier transmission.

The population-level competition in our between-population selection is expected to be effective at promoting cooperation in that it replicates a process akin to 'budding dispersal': successful lineages divide in each passage and are bottlenecked through single aphid infections [61,62]. Such means of dispersal preserves population structure between selected lineages and

has the potential to increase relatedness and decrease conflicts [61,62]. Put another way, bottle-necking infections through single aphids should reduce the genetic variation within sub-populations in a lineage and maximizes genetic variation between subpopulations making selection at population level more efficient. In line with this expectation, the between-population selection lines had fewest genetic variants within lineages, while the treatment that pooled infections from multiple cadavers (between-host) retained the highest number of variants relative to other evolved lineages.

The other two selection regimes also aimed to manipulate levels of selection but generated more inconsistent changes in phenotype. The pooling of cadavers in order to select for high yield (between-host treatment) produced some unexpected challenges since bacterial contamination led to the extinction of multiple lineages. It seems likely that disruption of the integrity and spatial structure of the mycosing cadaver provides an opportunity for bacterial growth that is suppressed in intact cadavers. It is well established that the benefits of antimicrobial secretion are optimized in strongly structured environments [63,64], the negative consequences of disrupting this structure suggest quite intense fungal-bacterial competition in infected hosts. Irrespective of the cause, aphid pooling treatment produced some of the most dramatic shifts in phenotype—late sporulation, low colony growth and also one lineage with high virulence. These changes were accompanied by mutations in a number of putative regulatory genes (i.e. transcription factors and zinc finger proteins) which are capable of generating strong phenotypic differences with little genetic change.

Extinction also occurred in half of the within-host treatment replicates. Here we attempted to increase the strength of within-host competition by diminishing the importance of between-host and between-population competition. A clear result from this treatment was that it did not prove possible to alter speed kill despite specifically selecting for early death. Across the experiment as a whole, it proved very hard to change time to death any earlier than 5 days, suggesting some hard constraints to fungal development in cadavers. Only the high virulence between-host treatment lineage BHY3 produced substantial death at day 4, and this trait was accompanied by poor growth and sporulation characteristics. We anticipated that strong within-host competition might lead to cheat invasion and extinction—this was one reason we set up eight replicates in this treatment. The extinction of four out of these eight replicates because of failure to infect is consistent with invasion of low virulence mutants or cheats, but hard to confirm. One possible explanation for the within-host treatment producing lineages with decreased or increased infectivity is that narrowing of genetic diversity during single passage exposed different genotypes to selection across these eight replicates. We minimized between-host selection *within* replicates in the within-host treatment as selection occurred between-hosts in a single group (bioassay chamber) only. Nevertheless, across the whole within-host treatment we saw extinction of low virulence lineages and survival of lineages with improved sporulation or virulence and so an improvement in the mean virulence of survivors by the end of the experiments. This mimics at a reduced scale the selection between populations that we imposed in every passage of the between-population treatment- forced extinction of fungi low virulence groups and propagation of EPF from high virulence groups.

On the face of it, it may appear that differences between selection treatments were relatively subtle. Nevertheless, resequencing of evolved lineages showed convergent evolution at multiple loci across most experimental treatments confirming that the treatments had meaningful impacts on genetic variants within our starting population. Convergent evolutionary change, while common in experimental evolution in media [65–67] is, in our experience, not typical for *in vivo* passage experiments [19,68] and so this result is striking. The only treatment not to show convergent evolution was the media passage, where presumably there are multiple possible means of generating low sporulation efficiency genotypes that so commonly occur in repeated laboratory culture

[69–72]. The need to propagate fungi *in vitro* prior to infection means that there might be alternating selection acting against adaptation to the *in vivo* environment. Previous studies have shown that this does not prevent meaningful adaptation to hosts [19,68]. The media selection treatment was set up to control for any confounding effects. Furthermore, the ability to grow and sporulate well *in vitro* is also key for successful development of EPF as biocontrol agents, so retaining selection for efficient growth on media could be valuable.

The highest number of convergent SNPs were identified within protein coding sequences of transcription factors and thus may play a role in signalling pathways and regulation of gene expression in fungi [73,74]. Mutations in regulatory genes are one means of generating substantial changes in life history with limited genetic change and were one of the most common class of mutation in passage experiments with entomopathogenic bacteria [19]. The limited number of well-characterized transcription factors linked to stress and virulence pathways in entomopathogenic fungi make it hard to connect these specific proteins with their respective functions [75]. Nevertheless, a number of transcription factors have been shown to contribute to growth, conidiation and stress response in *B. bassiana* and *M. robertsii* [76] as well as chitinase activity [77] and cuticle penetration in *Metarhizium robertsii* [78]. Genomic variants of different transcription factors showing signs of convergent evolution were found in each of the *in vivo* selection treatments, suggesting that these transcription factors are involved in pathways that are advantageous *in vivo* but not *in vitro*. We identified two variants in the within-host selection treatment that have been associated with specific metabolic processes (lipid, glucose) in yeast [49,55]. Transmembrane activity (methionine transport) in the within-host treatment and an ABC efflux transporter in the between-population treatment were also associated with convergent genetic variants. ABC transporters are membrane-associated proteins that are involved in biological processes ranging from fundamental cellular processes to detoxification [51,79,80] and have been shown to affect virulence in *B. bassiana* [81]. Although available genomic resources for *Metarhizium spp.* and *Beauveria bassiana* strains have been increasing due to their commercial importance, there is still a lack of functional annotation for entomopathogenic fungi in general, and *A. muscarius* in particular, that would be key to improving our understanding of the mechanistic basis for convergent variants that evolved, in this study.

Aside from speed of kill, this experimental evolution demonstrated that there is considerable selectable variation and genomic flexibility in fungal entomopathogens [82–84]. This is especially noteworthy as our starting population was a commercial strain. Commercial biocontrol agents are typically repeatedly subcultured or passaged as single colony variants to minimize genetic diversity and preserve trait stability [85]. Importantly, we found no consistent trade-offs between gains in virulence, and radial growth or sporulation efficiency, in contrast to theory and previous experience with entomopathogens [19]. Efficiency of sporulation and timing of sporulation are extremely important for cost-effective biocontrol agent production, and in this study, it proved possible to improve both sporulation and virulence traits simultaneously. Gains in virulence were also common to multiple aphid species and were not the result of specialization on the experimental host. This is particularly encouraging for the application of experimental evolution to strain improvement in fungi. A caveat here is that sporulation efficiency in laboratory conditions does not necessarily translate into commercial production [30,44]. Nevertheless, that is something that could be addressed in future experiments.

## 4. Methods

### Ethics statement

Ethical approval for this research was gained via the Faculty of Environment, Science and the Economy Cornwall Ethics Committee appropriate for invertebrates, application 524112.

## Insect and fungal culture

Populations of *M. persicae* and *Brevicoryne brassicae* were reared in BugDorm-4 Polyester Mesh rearing cages (NHBS, UK) on Chinese cabbage, *Brassica pekinensis* plants at BBCH growth stage 13–15. Colonies were sub-cultured as required by transferring 5–15 apterous adults onto fresh plants. Individuals of defined age were produced by confining cohorts of mature apterous virginoparae in clip cages for 48 hours to produce progeny. Subsequently, adults were removed, and the nymphs maintained for further nine days. All insects were maintained at 24±1°C, 14L: 10D photoperiod, which ensured the maintenance of an anholocyclic (asexual) life cycle. *M. persicae* clone 'O' was originally collected from potato plants in the UK in 2007, *Brevicoryne brassicae* clone 'K3' was obtained from Brussels sprouts in the UK in 1997 and supplied by the Warwick Crop Centre.

The ancestral strain of *A. muscarius* was isolated from Mycotal (Koppert, NL) and stored on porous beads in cryovials at -80°C. One bead was removed in sterile conditions to inoculate an SDA agar slant that after 10 days of growth at 23°C was kept in 4°C for roughly 2 months. Spore suspensions for selection and specific assays were obtained by using a sterile loop to inoculate SDA agar plates from this agar slant. SDA plates were incubated in darkness at 23 ±1°C for 10–14 days. This two-step culturing protocol aims to minimize the alteration of fungal phenotypes by subsequent culturing in vitro [86]. Evolved lineages were stored and cultured as described above. Conidial suspensions were prepared by agitating mycelia in SDA plates with 'L-shaped' spreaders after the addition of 10 ml of 0.01% v/v Triton X-100. The suspensions were then passed through sterile cheesecloth to remove any hyphal fragments. Suspensions were enumerated using Fastread102 counting chambers (Kova International, USA) and adjusted to the required concentration in sterile water with 0.01% Triton X-100. This method was used for all enumeration of conidia.

## Optimization of UV-C mutagenesis

Conidial suspensions of known concentration were prepared as described above. Random UV mutagenesis was carried out in a biological safety cabinet, utilizing the integrated disinfection UV lamp (36W) at a distance of 74 cm from the plates. The radiation of the lamp was stabilized by switching it on 30 min prior to exposure. Aliquots of 10 ml spore suspensions were exposed to UV-C radiation for 0, 45, 60 and 90 seconds in 90 mm Petri dishes without lids and 100 μl aliquots were spread on SDA plates followed by incubation at 23°C for 5 days. Treatments were carried out with three replicates. Control and 60 seconds treatments of the experiment were additionally repeated at another time. Conidial lethality ratios were calculated as: (1 – (number of colonies on radiation-treated plates/ number of colonies on control plates)) x 100%. Conidial lethality rates increased with prolonged exposure time. Exposure for 45, 60 and 90 seconds resulted in mean conidial lethality of 33 ± 5.7, 50.4 ± 1.7 and 82.9 ± 1.7% respectively.

## Selection protocol

Cohorts of 13 even-aged young adults of *M. persicae* were split into three treatments: between-population selection for infectivity, within-host selection for speed of kill and between-host selection for yield (referred to as: between-host, within-host and between-host selection respectively). All treatments were initially infected with the ancestor *A. muscarius*. The between-population treatment had four replicates with four subpopulations in each replicate, while the within-host and between-host treatments were initiated with eight replicates.

The cohorts of 13 aphids were sprayed with 400 μl of conidia suspension using a precision micro sprayer in a 55 mm Petri dish as described by Erdos, Halswell [87]. After treatment,

aphids were maintained on single leaves of Chinese cabbage in plastic cups with mesh covered vents at 20±1˚C and a 14:10 L:D regime. Nymphs were removed daily. Dead insects were surface sterilized by dipping in 70% ethanol and rinsed in sterile distilled water followed by drying on filter paper before being placed on SDA as intact cadavers for production of conidia as per methods above.

Selection was applied as follows: the between-population selection treatment was propagated from the first aphid cadaver belonging to the most successful subpopulation (highest % mortality at 7 days past infection) in a method adapted from previous work [19]. In the case of multiple subpopulations with the same proportional mortality, selection was based on the fastest speed of kill. The remaining three subpopulations were discarded. The within-host selection treatment was passaged from the first aphid to succumb to fungal infection in each replicate, while the between-host selection treatment was propagated from the first 3–4 mycotsed aphid cadavers, (i.e. those with emergent sporulating mycelium): these were homogenized in sterile 0.01% V/V Triton X-100 in an Eppendorf with a pellet pestle. All sterilized cadavers and aphid homogenates were left to sporulate on SDA plates as per methods above (Fig 1). An additional control treatment was carried out to study the effect of adaptation to growth media by passaging the ancestral strain of *A. muscarius* on SDA plates only. Inoculation of fresh plates used plugs of conidia and hyphae taken with a sterile 9 mm diameter cork borer after 10–14 days of growth and used three independent replicates (Fig 1).

Seven rounds of passage were carried out for all treatments. In order to increase genetic variation, passage rounds 2, 4 and 6 included a step of random mutagenesis before infecting the next cohort of aphids. Random mutagenesis was carried out by exposing fungal spore suspensions to UV irradiation for 60 seconds resulting in 50% germination, as determined previously. The applied dose was adjusted after mutagenesis to compensate for conidia mortality.

In each round of *in vivo* selection, we aimed to impose mortality averaging between 25–50% across all treatments. We used a standardized dilution factor across each treatment based on the conidia counts of a randomly subsampled lineage in each passage round. This dilution factor varied between passage rounds to correct for changes seen in percentage mortality over time. Mortality rates were maintained within our target range, except when selected lineages went extinct.

## Measuring changes in pathogen life history traits after selection

After the selection rounds were completed, we conducted a range of phenotypic assays to assess changes in fungal life history. Here we focused on the life history parameters that we predicted would be altered by the different selection regimes.

## Infectivity, speed of kill and virulence

In order to measure changes in virulence, 18–20 adult apterous aphids were sprayed in a 55 mm Petri dish with 400 μl of $3 \times 10^6$ conidia/ml suspension of selected lines of *A. muscarius* (87). After treatment, aphids were maintained on single leaves of Chinese cabbage in plastic cups with mesh covered vents at 20±1˚C and a 14:10 L:D regime. Aphid mortality was recorded daily for 7 days post-treatment. Nymphs were removed daily. Dead insects were surface sterilized with 70% ethanol and rinsed in sterile distilled water. Sterilized cadavers were plated on SDA agar and observed for fungal outgrowth to confirm death was caused by fungal infection. Bioassays of lineages were carried out with five replicates. Controls were treated with a sterile carrier (0.01% Triton-X-100). In order to test if changes in virulence were specific to our passage host, or more general for aphid infection, we repeated the bioassays for a subset of

lineages in an additional aphid host, *B. brassicae* using a dose of $1x10^7$ conidia/ml with four replicates.

## Mycelial growth rate

We used radial growth on agar plates to measure changes in *in vitro* growth rate following selection. Radial growth of the ancestor and selected lineages were assessed by inoculating SDA plated in 90 mm triple vented Petri dishes. Each Petri dish was split into four equal parts and inoculated in the center of the quadrant with 7 μl of a $3x10^6$ conidia/ml suspension. Experiments were replicated three times and plates incubated at 22±1˚C in darkness for 8 days. Radial growth was measured daily with two cardinal diameters drawn on the bottom of each plate. Radial growth was measured every day (except day 9 and 10) by two cardinal diameters drawn on the bottom of each plate.

## Spore production

The rates and quantities of spore production of the ancestral and evolved lines were assessed *in vitro* at 3 and 14 days post inoculation. SDA plates were inoculated by spreading 100 μl of $1x10^7$ conidia/ml suspension on each plate. After 3 days of incubation a sample of each plate was taken with a 9 mm diameter sterile cork borer, suspended and vortexed in 1 ml of 0.01% V/V Triton X-100 in a 2 ml Eppendorf tube to recover spores. Final spore production after 14 days of incubation was assessed by agitating mycelia with a 'L-shaped' spreader in 10 ml of 0.01% V/V Triton X-100. Conidia were enumerated as described above and each lineage was assessed in three independent repeats.

## Sequencing and variant discovery

Long-read sequencing and library preparation for the ancestor *A. muscarius* was carried out as previously described [88]. For short-read re-sequencing of ancestral and evolved lineages we harvested mycelia from 14-day-old fungal cultures grown on SDA plates in 22±1˚C in darkness. DNA was extracted using the QIAGEN Genomic-tip kit and quantified using a Nano-Drop spectrophotometer 2000 (Thermo Scientific, Loughborough, UK) and Qubit Fluorometer using the Qubit dsDNA BR assay kit (Thermo Scientific, Loughborough, UK). DNA integrity was checked using agarose gel electrophoresis. We generated Illumina libraries for all derived lineages and the ancestor using the Illumina NEBNext Ultra II FS Kit, that was sequenced on a single lane of Illumina NovaSeq SP, resulting in 44.5 million pairs of 150-bp with an average insert size of 210 bp.

Variant discovery was carried out using the Genome Analysis Toolkit (GATK) pipeline v4.1.0.0 to call both variant and reference bases from alignments [89]. Prepared Illumina libraries of the evolved and ancestral lineages were mapped to the previously assembled *A. muscarius* genome [88] with BWA v2.2.1 [90,91]. Alignment pre-processing was carried out using Picard tools AddOrReplaceReadGroups, MarkDuplicates (http://broadinstitute.github.io/picard/), followed by indexing with SAMtools v1.10 [92]. GATK HaplotypeCaller was used to call variants with the haploid setting, followed by CombineGVCFs, GenotypeGVCFs to perform joint genotyping. The minimum phred-scaled confidence threshold at which variants should be called was set at the default value of 30.0. Filtering of variants was carried using the parameters "QD < 2.0", "MQ < 40.0", "FS > 200.0" using VariantFiltration. Bases with a read depth of < 50 or > 8000 were removed, as well as sites not including the GATK PASS flag. Genetic differentiation was analysed using vcfR [93]. High-quality variants that were fixed in more than half of the individually evolving replicates were considered to be convergent mutations.

### In vitro competition and quantitative PCR

*In vitro* competitive fitness of lineages showing improved early sporulation and virulence belonging to the between-population treatment (BPI1, BPI3, BPI4) were tested against a media selected lineage. In the sequencing and variant discovery work above, we identified a SNP present only in lineage Media1 (Chromosome 1, position 4527201, G/A) and designed a Custom Taqman SNP Genotyping Assay (Thermofisher, UK) that can detect this specific SNP using a FAM-labelled probe for this SNP and a VIC-labelled probe for the reference base. Using this SNP as a marker, we competed BPI1, BPI3, BPI4 and the ancestor pairwise against Media1 by combining 50 μl of $3\times10^6$ conidia/ml of each pairwise combination and plating 30 μl on the center of SDA plates. Each combination was repeated three times and incubated at $22\pm1°C$ in darkness for 14 days. All fungal matter was extracted from the agar plates using a sterile scalpel, and snap frozen in liquid nitrogen. DNA was extracted using the Dneasy Plant Mini kit (Qiagen, UK) following the manufacturers' instructions. Extracted DNA was quantified with a Qubit Fluorometer using the Qubit dsDNA BR assay kit (Thermo Scientific, Loughborough, UK) and concentrations adjusted to 20 μg/ml.

Each qPCR reaction consisted of 12.5 μl Taqman Master Mix (Thermofisher, UK), 1 μl of probe mix and 1 μl DNA. Temperature cycling comprised 10 minutes at 95°C followed by 40 cycles of 95°C for 15s and 60°C for 60s. Each qPCR reaction was run with three independent biological replicates and three technical replicates. Relative fitness was derived from relative fluorescence units obtained via qPCR and calculated as $v = x_2(1-x_1)/x_1(1-x_2)$, where $x_1$ is the initial proportion of ancestral allele (present in both ancestor and between-population treatments) in the population and $x_2$ is their final proportion. The value of $v$ signifies whether the ancestral allele increases in frequency ($v > 1$), decreases in frequency ($v < 1$), or remains at the same frequency ($v = 1$) as described in [94]. Comparisons of $v$ were made following log transformation.

### Statistical analysis

Data analyses were carried out in R version 3.6.0 [95]. Conidia viability after mutagenesis was analyzed using one-way ANOVA, comparisons were carried out using Tukey HSD with 95% confidence intervals. Overall virulence was modelled using survivorship analysis with Cox proportional hazard models ('Survival') [96]. Generalized linear models (GLM) with binomial error structure and a logit link function were used to analyze infectivity (total mortality in bioassays), treatment as fixed effects, experimental block as random effect. Speed of kill was analyzed as simple untransformed time to death in GLMs, this gave qualitatively similar results to models with Gamma errors. Pairwise comparisons were made using Tukey HSD with 95% CIs. Where model comparisons could not be made using F tests, due to the models having the same degrees of freedom, we compared explanatory power using the Aikaike Information Criterion (AIC) [97]. The relationship between colony radius and time was assumed to be linear. Colony growth rate was estimated from the slope of linear regression of colony radius on time. Pairwise combinations between the ancestor and different selection treatment on radial growth was assessed comparing marginal means using Tukey's HSD. All spore counts data were square root transformed; analyses of response to selection treatments were fitted to a linear mixed effects model in *lme4* with independent lineage as a random effect [98].

### Dryad DOI

https://doi.org/10.5061/dryad.9s4mw6mpz [99]

## Supporting information

**S1 Fig. Violin plot of time to death of evolved and ancestral lineages.** Red dots represent the mean with standard errors.
(TIF)

**S2 Fig.** Total mortality of *B. brassicae* after 7 days of exposure to selected lines of *A. muscarius* (A). Boxplots showing median, first and third quartiles, whiskers are 1.5 * interquartile range (IQR). Data beyond the end of the whiskers are outliers. Mean survivorship curves of *B. brassicae* over 7 days after treatment with selected lines of *A. Muscarius* (B). Significance levels for survival model: ***: $p < = 0.001$; *: $p < = 0.05$.
(TIF)

## Acknowledgments

The authors would like to acknowledge the use of the University of Exeter's Advanced Research Computing facilities in carrying out this work. This project utilised equipment funded by the Wellcome Trust (Multi-User Equipment Grant award number 218247/Z/19/Z).

## Author Contributions

**Conceptualization:** Zoltan Erdos, Ben Raymond.

**Data curation:** Zoltan Erdos, Manmohan D. Sharma.

**Formal analysis:** Zoltan Erdos, Ben Raymond.

**Funding acquisition:** David Chandler, Chris Bass, Ben Raymond.

**Investigation:** Ben Raymond.

**Methodology:** Zoltan Erdos, David J. Studholme, Ben Raymond.

**Project administration:** Zoltan Erdos.

**Resources:** Manmohan D. Sharma, David Chandler, Chris Bass, Ben Raymond.

**Software:** Zoltan Erdos, David J. Studholme, Manmohan D. Sharma.

**Supervision:** David Chandler, Chris Bass, Ben Raymond.

**Validation:** Zoltan Erdos, Manmohan D. Sharma.

**Visualization:** Zoltan Erdos.

**Writing – original draft:** Zoltan Erdos, Ben Raymond.

**Writing – review & editing:** Zoltan Erdos, David J. Studholme, David Chandler, Chris Bass, Ben Raymond.

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
