## [Decision Letter · Decision Letter 0]

18 Jan 2024

Dear Dr Erdos,

Thank you very much for submitting your manuscript "Manipulating multi-level selection in a fungal entomopathogen reveals social conflicts and a method for improving biocontrol traits" for consideration at PLOS Pathogens and we apologise again for the delay in processing this manuscript. As with all papers reviewed by the journal, your manuscript was reviewed by members of the editorial board and by several independent reviewers. The reviewers appreciated the attention to an important topic. Based on the reviews, we are likely to accept this manuscript for publication, providing that you modify the manuscript according to the review recommendations.

The reviewers and editorial board found this manuscript of great interest but there were a number of concerns that we felt must be addressed. These are highlighted in at attached reviews but we would like to emphasis the comments made by reviewer 2. The manuscript needs to be revised so that it is clear how the terminology used relates tot he experiments conducted. Terminology must be consistent and, as much as possible, simple and self-explanatory. When submitting your revision please ensure that you provide a marked up copy of the changes made to address the reviewers' concerns (see point 2 below).

Sincerely,

Alex Andrianopoulos

Section Editor

PLOS Pathogens

Alex Andrianopoulos

Section Editor

PLOS Pathogens

Michael Malim

Editor-in-Chief

PLOS Pathogens

orcid.org/0000-0002-7699-2064

Reviewer Comments (if any, and for reference):

Reviewer's Responses to Questions

**Part I - Summary**

Reviewer #1: This is an interesting research article on multi-level selection in a fungal entomopathogen to understand the trade-offs in life history traits while trying to increase virulence. Overall, the research has merit but there are a few concerns that must be addressed. 1) The authors claim that virulence of fungal parasites typically fails to increase when passed through their insect host. However, this claim is contrary to what the literature suggests. Research has shown that passage of an entomopathogenic fungus through its insect host increases fungal virulence. Please acknowledge this information and then provide the gap in literature you are trying to fill (Example paper: Evison, et al. 2020. J. Evol. Bio. 28:179-188). 2) Typically, experimental evolution studies do not involve several rounds of UV mutagenesis. Since UV mutagenesis was performed, the results should be interpreted with caution. For example, in lines 477-481, the genomic flexibility of the commercial strain of Akanthomyces muscarius is highlighted, but the observed genomic variation could be a result of UV mutagenesis. 3) Considering that Akanthomyces muscarius is a filamentous fungus, what is the biological interpretation of the social evolution theory? In filamentous fungi, cells are not discrete entities but are interconnected. So, the occurrence of cheater cells needs a more biologically relevant explanation. For example, heterokaryosis is commonly known to occur in filamentous fungi in which two or more genetically distinct nuclei are present in the same cell (Gambhir, et al. 2022. Msphere, 7:e00087-22). Please see additional comments below.

Reviewer #2: In their manuscript entitled Manipulating multi-level selection in a fungal entomopathogen reveals social conflicts and a method for improving biocontrol traits, Erdos et al. use serial-passaging to try and reveal trade-offs in key pathogen traits important for infectivity and virulence. Interestingly the authors do not discover any of the normally predicted trade-offs in obligate-killing pathogens, but describe a phenotypic change to earlier sporulation in a between group selection treatment. The findings are novel and exciting by bringing and using social evolution theory to entomopathogenic fungi, and as such should be of interest to a wide audience.

The manuscript is well written and figures easy to follow. Apart from some minor comments listed below, my main concern with this manuscript is that the use of developed terminology is confusing, and makes the introduction and the links to the experimental plan difficult to follow. For example, when referring to between group treatment in the abstract, is that then the between populations or between host treatment, I understood the former but it was not perfectly clear. Furthermore, the three different treatments are referred to as speed of kill treatment, infectivity selection, and yield treatment, but the link to the different scales of selection described in the introduction as: between populations, between hosts, and within hosts are not easy to follow. The different names of the treatments are introduced at the end of the introduction, but should be more clearly defined with links to the scale of selection (i.e. between, within hosts or between populations). When these terms are used in the first paragraph of the results section on page 10, it is not immediately clear what the different treatments involve. As a reader, I found figure 7 very informative, and I suggest to re-organize the manuscript to introduce figure 7 much earlier, maybe as a new first figure and specifying the links to the scales of selection, so the experimental set-up is more easily linked to introduction. For example, on L192-193 it is stated that: “Total mortality is sometimes described as infectivity, but we will avoid this term to prevent confusion with the infectivity selection regime.”. But at this point in the text, the infectivity selection regime is not clearly defined. I assume this is defined on L149-151, but to make it more clear for readers when defining these experiments this sentence could start with: “Infectivity selection regime: We predicted … etc.” or similar to make it unambiguous for readers.

If the methods were presented before the results section, the manuscript would be much easier to follow, but since this is not the structure of this journal, care must be taken to make it comprehensible for readers to go straight from the introduction to results. There are numerous examples of details referred to in the results that have not been properly introduced, for example on line 205 where specific lineages are mentions, BPI2 etc., but what does BPI stand for? This is only mentioned in the figure legend. One way to remedy this is to add a new first paragraph (maybe including a reference to a new figure 1 as mentioned above) at the start of the results, providing more detail of the experimental set-up. Alternatively, the final paragraphs of the introduction could be reworked to make the experimental set-up more easily comprehensible.

If I understand the experimental protocol correctly, the infected and chosen cadavers were collected, surface sterilized, and placed on SDA agar. After fungal growth and sporulation on the agar, conidia were harvested and used to inoculate the next round. This imply that the fungi effectively alternates between growing on and inside an aphid and then on agar in each round. I perfectly sympathize with the authors for this methodology, but would like to see a discussion of this fact incorporated in the manuscript, as this may have an effect – especially on the strength of selection imposed by the Aphid environment.

**Part II – Major Issues: Key Experiments Required for Acceptance**

Reviewer #1: (No Response)

Reviewer #2: No new experiments required

**Part III – Minor Issues: Editorial and Data Presentation Modifications**

Reviewer #1: - Line 26. As explained in point 1), this is contrary to what the literature suggests.

- Line 29. This sentence needs to be modified to ensure accuracy. The experiment was not started with a genetically diverse stock, but with the ancestor A. muscarius. To increase genetic variation, random mutagenesis was induced later.

- Lines 90-98. Since this paper is focused on fungal enothomopathogens, virulence should be discussed in this context instead of solely relying on entomopathogenic nematodes. There is ample of evidence that suggests that passage of an entomopathogenic fungus through its insect host increases fungal virulence. Please acknowledge this information and then provide the gap in literature you are trying to fill.

- Lines 100-115. Although this is a good explanation of the social evolution theory. Considering that Akanthomyces muscarius is a filamentous fungus, what is the biological interpretation of the social evolution theory? Please refer to point 3) above for more details.

- Line 253. Uppercase 'I' should be replaced with lowercase in 'I.e'

- Line 513. What is the required conidial concentration?

- Lines 530-531. Please mention the within-host selection.

- Figure 3 B. The yellow plus signs are not visible. Please consider increasing the thickness of the symbol and/or changing the color.

- Lines 477-481. The observed genomic variation could be a result of UV mutagenesis.

Reviewer #2: Minor comments:

L54 – dependent on how?

L164 - While several?

L201 – Please consider to provide the phenotypic effect sizes.

L247 – Could significantly higher early sporulation be replaced with significantly earlier sporulation?

L277 - Figure 3 – Is it possible to add which linear model estimates are significantly different from the ancestor as indicated on L269?

L319 – How many SNPs were inside or in close proximity to coding regions? How were the candidate SNPs presented in Table 1 chosen?

L359-360 – I did not read the introduction as this being your primary hypothesis, please consider adjusting in line with my main comment above regarding the use of developed terminology and presentation of hypotheses and experimental design.

L379 – cells that are sporulating?

L385-387 – How does this reduced competition play out if the early sporulating variant has a head-start and has grown to a bigger biomass when the late sporulation variant enters the niche? Does this change the trajectory of the degree of the tragedy of the commons?

L418 – Which regulatory genes?

L507 – How many transfers of the fungal isolate on media was performed prior to the experiments?

L578 – after the experiments, were the fungal isolates isolated and preserved for long-term storage in a freezer or maintained on agar, and if so for how many transfers before being assessed for changes in pathogen life history traits? Please add a bit more information on how the ancestor and selected lines were maintained (or immediately used) to assess phenotypic changes, as just a few in-vitro transfers on agar can alter the phenotype of fungi.

PLOS authors have the option to publish the peer review history of their article (what does this mean?). If published, this will include your full peer review and any attached files.

Reviewer #1: No

Reviewer #2: No

Figure Files:

Data Requirements:

Reproducibility:

References:

---

## [Editor Report · Decision Letter 1]

9 Mar 2024

Dear Dr Erdos,

We are pleased to inform you that your manuscript 'Manipulating multi-level selection in a fungal entomopathogen reveals social conflicts and a method for improving biocontrol traits' has been provisionally accepted for publication in PLOS Pathogens.

Best regards,

Alex Andrianopoulos

Section Editor

PLOS Pathogens

Michael Malim

Editor-in-Chief

PLOS Pathogens

orcid.org/0000-0002-7699-2064
---

## [Editor Report · Acceptance letter]

21 Mar 2024

Dear Dr Erdos,

We are delighted to inform you that your manuscript, "Manipulating multi-level selection in a fungal entomopathogen reveals social conflicts and a method for improving biocontrol traits," has been formally accepted for publication in PLOS Pathogens.

Best regards,

Michael Malim

Editor-in-Chief

PLOS Pathogens

orcid.org/0000-0002-7699-2064